# Comparative Analysis of Different Proteins and Metabolites in the Liver and Ovary of Local Breeds of Chicken and Commercial Chickens in the Later Laying Period

**DOI:** 10.3390/ijms241814394

**Published:** 2023-09-21

**Authors:** Yuan Tang, Lingqian Yin, Li Liu, Qian Chen, Zhongzhen Lin, Donghao Zhang, Yan Wang, Yiping Liu

**Affiliations:** Farm Animal Genetic Resources Exploration and Innovation Key Laboratory of Sichuan Province, Sichuan Agricultural University, Chengdu 611130, China; 2021202030@stu.sicau.edu.cn (Y.T.); yinlingqian@stu.sicau.edu.cn (L.Y.); liuli55@stu.sicau.edu.cn (L.L.); 2021302176@stu.sicau.edu.cn (Q.C.); zzlin599@163.com (Z.L.); 2019202006@stu.sicau.edu.cn (D.Z.); wangyan519@sicau.edu.cn (Y.W.)

**Keywords:** aged hens, proteomics, metabolomics, UHPLC-MS/MS, reproductive performance

## Abstract

The liver and ovary perform a vital role in egg production in hens. In the later laying period, the egg-laying capacity of female hens, particularly that of local breeds, declines significantly. Hence, it is essential to study the features and conditions of the ovary and liver during this period. In this research, we characterized the proteins and metabolites in the liver and ovary of 55-week-old Guangyuan gray chickens (Group G) and Hy-Line gray chickens (Group H) by using liquid chromatography chip/electrospray ionization quadruple time-of-flight/mass spectroscopy (LC-MS/MS). In total, 139 differentially expressed proteins (DEPs) and 186 differential metabolites (DMs) were identified in the liver, and 139 DEPs and 36 DMs were identified in the ovary. The upregulated DEPs and DMs in both the liver and ovary of Group G were primarily enriched in pathways involved in amino acid and carbohydrate metabolism. This suggests that energy metabolism was highly active in the Guangyuan gray chickens. In contrast, the upregulated DEPs and DMs in Group H were mainly enriched in pathways associated with lipid metabolism, which may explain the higher egg production and the higher fatty liver rate in Hy-Line gray hens in the later laying period. Additionally, it was found that the unique protein s-(hydroxymethyl) glutathione dehydrogenase (ADH4) in Group G was implicated in functions such as fatty acid degradation, glycolysis, and pyruvate metabolism, whereas the unique proteins, steroid sulfatase (STS), glucosylceramidase (LOC107050229), and phospholipase A2 Group XV (PLA2G15), in Group H were involved in the metabolism of steroid hormones and glycerol phosphate. In conclusion, variations in how carbohydrates, lipids, and amino acids are processed in the liver and ovary of local breeds of chicken and commercial hens towards the end of their laying period could explain the disparities in their egg production abilities.

## 1. Introduction

Poultry play a vital role in providing food for humans, and the reproductive performance of poultry directly impacts the development and benefits of the poultry industry. The utilization cycle of commercial chickens has been extended from 72 to 80 weeks as a result of enhancements in egg production capabilities [1], and Hy-Line chickens still maintain an egg production rate of about 80% at 53 weeks of age [2]. However, the egg production performance of laying hens declines significantly in the later laying period, which has a considerable impact on economic profitability. For the purpose of reducing breeding costs and increasing the utilization effectiveness, extending the egg-laying cycle with better reproductive performance of hens during the later laying period is a crucial task.

Previous research on animals’ reproductive performance has primarily concentrated on studying the ovary and liver. In the later laying period, the ovary and liver of chickens undergo changes in their structure and declines in their function, which subsequently impact their reproductive capacity [3]. The proper development of follicles in the ovary is crucial for egg production, and this process is regulated by the hypothalamic–pituitary–gonad (HPG) axis [4,5]. As chickens reach the later laying period, the release of sex hormones decreases, the number of pre-grade follicles decreases, and yolk synthesis and accumulation reduces, ultimately resulting in a decline in ovarian function [6]. The liver has multiple functions in poultry. It plays a major part in fatty acid metabolism and is responsible for producing the building blocks needed for yolk formation [7]. Estrogen (E2) controls this process and ensures that the follicles receive the necessary nutrients [8]. However, laying hens often accumulate excessive amounts of fat due to continuous egg production and high-energy diets during the later stages of laying [3]. This often leads to liver failure [9] and lipid metabolism disorders [10,11] in laying hens. Consequently, the overall health and egg production performance of the poultry during the later stages of laying can be adversely affected.

Previously, researchers have focused on improving the reproductive performance of later laying hens through feed additives to improve the synthesis of hormones [12,13], eggshell quality [14], and antioxidant capacity [15,16,17]. With the development of technologies such as the transcriptomics, many genes and transcription factors have been identified that affect the reproductive performance of poultry during the later laying period. For example, *ACSF2*, a gene involved in lipid metabolism, was highly expressed in the liver of high-producing chickens [18]. However, transcriptomic information does not fully explain the complexity and dynamics of the regulation of reproduction in poultry. Proteomics and metabolomics, as new technologies in the post-genomic era, provide new insights into and means to understanding the molecular changes in reproduction in poultry. Proteomics and metabolomics are now widely used in avian breeding, such as in screening for key growth factors by evaluating differently expressed proteins (DEPs) and differential metabolites (DMs) in the pectoral muscle and intramuscular fat of chickens with contrasting growth rates [19], investigating the mechanisms of the functional regulation of granulosa cells in poultry by identifying DEPs and DMs in the ovary of laying chickens before and after sexual maturity [20], and screening for disease biomarkers of laying hens with fatty liver hemorrhagic syndrome (FLHS) by proteomics [21].

Until now, few studies have been reported on key regulatory proteins and metabolites in the liver and ovary of later laying chickens. The Guangyuan gray chicken is a high-quality meat and egg chicken in China. However, as it is a local breed, its egg production rate is only 80% during the peak laying period, and it mostly stops laying eggs at 55 weeks of age. In this study, to investigate the DEPs and DMs in the liver and ovary of local breeds of chicken and commercial laying hens during the later laying period, liver and ovary tissues from Guangyuan gray chickens (Group G) and Hy-Line gray chickens (Group H) were collected for proteomic and metabolomic analyses to distinguish the different proteins and metabolites in the liver and ovary of different chicken breeds during the later laying period. This research will lay the foundation for improving the performance of Guangyuan gray chickens in the later laying period.

## 2. Results

### 2.1. Comparison of the Morphological and Histological Characteristics of the Liver and Ovary between Guangyuan Gray Chickens and Hy-Line Gray Chickens

Body weight, liver weight, ovarian weight, oviduct length, and follicle number were collected from Guangyuan gray chickens (Group G) and Hy-Line gray chickens (Group H). As shown in Table 1, compared to Group H, Group G had a markedly lower body weight, liver weight, number of yellow follicles (YF, 5–10 mm), number of F1–F6 follicles, and oviduct length, but no apparent difference in counts of white follicles (WF, <5 mm).

According to the morphological and histological observations, the livers of Group G were smaller in size and reddish in color (Figure 1A), and HE staining showed that the cytoplasm of the hepatocytes in Group G was mildly stained, but the nuclei of the cells were darkly stained (Figure 1B). Oil Red O staining showed that the cellular interstitial space was narrow, and the distribution of lipid droplets was uniform (Figure 1C). In contrast, the livers from Group H were larger and yellowish in color (Figure 1D). The hepatocytes were neatly and clearly arranged, with a pink cytoplasm and light blue nuclei (Figure 1E), and the cytoplasm was sparse with large lipid droplets (Figure 1F). Observations of the ovaries showed that the ovaries in Group G contained only a small number of primordial follicles and pre-grade follicles (Figure 1G,H), and the ovaries were low in lipid droplets and had a low rate of yolk deposition in the follicles (Figure 1I). In contrast, Group H had ovaries containing more than four graded follicles (>12 mm), follicles filled with yolk and containing more than two layers of granulosa cells (Figure 1J,K), and ovaries with a high content of lipid droplets and a high rate of yolk deposition in the follicles (Figure 1L).

### 2.2. Identification of Differentially Expressed Proteins in the Liver and Ovary

Prior to the LC-MS/MS analysis, the quality of the samples was examined using BCA and SDS-PAGE, which showed that no degradation of the proteins occurred and that the amount of protein was sufficient for the subsequent experiments (Appendix A). The results revealed that in the liver, the number of spectra, peptides, and proteins identified in the non-redundant (NR) database and the Uniprot database were 140,517, 30,026, and 4167, respectively (Appendix A). The mass distribution of the proteins showed that the minority of proteins had a mass of 180–190 kDa, while the majority of proteins were distributed in the 20–30 kDa range (Appendix A). The results for the distribution of protein coverage showed that the majority of proteins had a peptide sequence coverage of 5–10% and the minority of proteins had a coverage of 90–95% (Appendix A). In the ovary, the number of spectra, peptides, and proteins identified was 159,983, 34,497, and 4964, respectively (Appendix A). The mass distribution and peptide sequence coverage of the proteins were the same as in the liver (Appendix A). A principal component analysis (PCA) of the proteins revealed a high degree of intra-group aggregation and significant inter-group separation, proving the reliability of the model (Figure 2A,B). The results of the DEP analysis showed that 139 DEPs were identified in the liver between Groups G and H, of which 41 were downregulated and 98 were upregulated (Figure 2C). In total, 139 DEPs were identified in the ovary, of which 73 were downregulated and 66 were upregulated (Figure 2D). A hierarchical clustering analysis showed that the significant DMs obtained from the previous analysis could effectively separate the two groups (Figure 2E,F). Notably, the volcano plots and hierarchical clustering analyses did not include unique proteins for both groups, whereas all subsequent functional analyses included unique proteins.

### 2.3. Functional Enrichment Analysis of Differentially Expressed Proteins in the Liver and Ovary

A GO analysis showed that DEPs in the liver were involved in several biological processes (BP), with more DEPs included in functions such as cellular processes, metabolic processes, and bio-regulation. Among the cellular components (CC) in which the DEPs were involved, more DEPs were included in functions such as cells, cell fractions, and organelles, while among the molecular functions (MF) in which the DEPs were involved, more DEPs were included in functions such as binding and catalytic activity (Figure 3A). The results of the GO analysis of DEPs in the ovary were consistent with those of the liver (Figure 3B).

The results of the KEGG analysis showed that the DEPs identified in the liver were enriched in 93 pathways. Among them, the upregulated DEPs were enriched in 46 and 13 unique pathways in the livers of Groups G and H, respectively. Among the unique pathways in Group G, four pathways were associated with amino acid metabolism, four pathways were associated with carbohydrate metabolism, and three pathways were involved in lipid metabolism. Among them, ALDH18A1, which is involved in lipid metabolism, and ADH4, which is involved in a variety of metabolic pathways, were unique proteins in Group G. In Group H, DEPs were enriched in two pathways related to lipid metabolism and in the pathway related to amino acid metabolism. All three DEPs were involved in tryptophan metabolism, with CYP1A4 being a unique protein in Group H (Table 2 and Appendix A). The DEPs upregulated in Groups G and H in the ovary were enriched in 36 and 33 unique pathways, respectively. In Group G, seven pathways were involved in amino acid metabolism and four pathways were involved in carbohydrate metabolism. Notably, ADH4 was also expressed in the ovary of Group G. It is involved in the fatty acid degradation and tyrosine metabolism pathways in the liver, and in the retinol metabolism, methane metabolism, pyruvate metabolism, and tyrosine metabolism pathways in the ovary. In Group H, in addition to the pathways related to amino acid metabolism and carbohydrate metabolism, three pathways associated with the lipid metabolism and oocyte meiosis pathways were also identified (Table 3 and Appendix A). In addition, STS, LOC107050229, and PLA2G15 were enriched in the lipid metabolic pathway in the ovary, and were also unique proteins in Group H.

### 2.4. Identification of Differential Metabolites in the Liver and Ovary

A comparison of the spectral overlap of the total ion chromatograms (TIC) of quality control (QC) samples was carried out and the results showed that the response strengths and holding times of the peaks were basically superimposed, suggesting that there was little alteration due to instrumental errors during the whole experiment (Appendix A). A partial least squares discriminant analysis (PLS-DA) was conducted on all metabolites identified in the positive and negative ion modes to assess the association between the metabolites’ expression and the samples’ class. In addition, an orthogonal PLS-DA (OPLS-DA) was used to modify the results of the PLS-DA (Appendix A).

In the liver, the results of the PLS-DA showed more significant intra-group aggregation and inter-group segregation of the samples, indicating that the model was reliable (Figure 4A,B). By using UHPLC-MS/MS, 1619 differential metabolites (DMs) with identifying information were found in Groups G and H in this study (*p* < 0.05). Chemical classification showed that the largest proportion of these metabolites was made up of organic acids and derivatives (28.474%), followed by lipids and lipid-like molecules (17.542%) (Figure 4C). The results of the analysis of DMs showed that 978 metabolites were upregulated and 1514 metabolites were downregulated in the liver between two groups (Fold change (FC) > 1.5 or FC < 0.67) (Figure 4D,E). According to the results of analyzing the significant DMs, 186 significant DMs with annotation information were identified in the liver between two groups, of which 63 DMs were upregulated and 77 DMs were downregulated. A correlation analysis showed that 119 DMs were identified as relevant in the positive ion mode and 67 DMs were identified in the negative ion mode (Figure 4F,G). In the ovary, the PLS-DA analysis also showed the high reliability of the model (Figure 5A,B). In the positive and negative ion modes, 1191 DMs with annotation information were identified, most of which belonged to lipids and lipid-like molecules (28.67%), followed by organic acids and derivatives (21.662%) (Figure 5C). The analysis of the DMs showed that 223 DMs were upregulated and 225 DMs were downregulated (Figure 5D,E). In total, 36 significant DMs with annotation information were screened, of which 17 DMs were upregulated and 12 DMs were downregulated. The correlation analysis showed that 14 DMs were identified to be correlated in the positive ion mode, while 22 DMs were identified to be correlated in the negative ion mode (Figure 5F,G).

### 2.5. Functional Enrichment Analysis of Differential Metabolites in the Liver and Ovary

The significant DMs identified in this study were functionally annotated with reference to the KEGG and HMDB databases, and DMs without annotation information were excluded. The results showed that the upregulated DMs were enriched in 42 and 14 unique pathways in the liver of Groups G and H, respectively. In Group G, six pathways were associated with amino acid metabolism and six pathways were associated with carbohydrate metabolism. In addition, three pathways were linked to the endocrine system and two pathways were linked to lipid metabolism. In contrast, in Group H, four pathways were associated with lipid metabolism, and DMs were also involved in amino acid metabolism and translation. Among them, the pathways related to amino acid metabolism in Group G mainly included valine, leucine, histidine, tyrosine, and phenylalanine, while the DMs related to amino acid metabolism in Group H were all involved in the tryptophan metabolism pathway (Table 4 and Appendix A). In addition, DMs upregulated in the ovaries of Groups G and H were enriched in six and eight unique pathways, respectively. In Group G, two pathways were linked to amino acid metabolism and two pathways were associated with carbohydrate metabolism. In Group H, four pathways were involved in carbohydrate metabolism and two pathways were involved in amino acid metabolism. Furthermore, the DMs were enriched in ABC transporters and biosynthesis of primary bile acid (Table 5 and Appendix A).

### 2.6. Comprehensive Analysis of DEPs and DMs in the Liver and Ovary

By comparing the unique pathways of DEPs and DMs enriched in the liver and ovary of Group G and H, the overlapping enrichment pathways of DEPs and DMs were screened. The analysis showed that in the liver of Group G, upregulated DEPs and DMs were co-enriched in the MAPK signaling pathway, glutathione metabolism, ferroptosis, pyruvate metabolism, glycolysis/gluconeogenesis, propanoate metabolism, histidine metabolism, and the degradation of valine, leucine, and isoleucine (Figure 6A). In contrast, in Group H, upregulated DEPs and DMs were co-enriched in the tryptophan metabolism pathway (Figure 6B). In the ovary of Group G, upregulated DEPs and DMs were co-enriched in the thiamine metabolism pathway (Figure 6C), while in Group H, upregulated DEPs and DMs were co-enriched in ABC transporters and the pentose phosphate pathway (Figure 6D).

### 2.7. Validation of DEPs Associated with Carbohydrate Metabolism and Lipid Metabolism Pathways in the Liver and Ovary

The biochemical indices in the serum of the two groups were determined. The results showed that the level of FSH in Group G was significantly lower than in Group H. The levels of the follicle stimulating hormone (FSH) (Figure 7A), luteinizing hormone (LH) (Figure 7B), lecithin (LEC) (Figure 7C), vitellogenin (VTG) (Figure 7D), very-low-density lipoprotein (VLDLy) (Figure 7E), total cholesterol (TC) (Figure 7F), and triglyceride (TG) (Figure 7G) were significantly lower than those in Group H. We also validated the protein expression levels of the DEPs. In the liver, the expression of ADH4 was significantly higher in Group G than in Group H, and it was hardly expressed in Group H. And the protein level of FADS1 was significantly higher in Group H than in Group G (Figure 7H,I). In the ovary, the protein levels of STS and HMGCS1 were significantly higher in Group H than in Group G, and STS was basically not expressed in Group G (Figure 7J,K). The protein expression trends of these DEPs were consistent with the results of proteomic analysis.

## 3. Discussion

The laying performance of local breeds of chickens differs considerably from that of commercial laying hens and remains so during the late laying period [22]. The liver is the major organ of energy metabolism and fatty acid metabolism [21], and yolk deposition in the follicle depends on the transport of lipids and proteins from the liver to the ovary [23]. In this study, the proteomic and metabolomic profiles were compared between the liver and ovary of 55-week-old Guangyuan gray chickens and Hy-Line gray chickens. Morphological and histological studies showed that the livers of Group H were yellowish in color, with more loosely arranged liver cells, and were richer in lipid droplets than those of Group G under the same feeding conditions. The altered morphology of liver cells in Group H may be attributed to the long-term absorption of large amounts of nutrients, such as TC and TG, by the liver in order to maintain a high egg production [23]. Eventually, the fat absorbed by the liver exceeds the transport capacity of the apolipoproteins, leading to fatty liver syndrome (FLS) [24]. The ovaries of Group G were basically atrophied, whereas grade follicles filled with yolk still existed in the ovaries of Group H. In addition, the serum levels of FSH and LH, which are related to follicular development [25]; the yolk precursors LEC, VTG, and VLDLy [26]; and TC and TG, which are related to steroid hormone synthesis and lipid deposition [27], were significantly higher in Group H. All the results indicated that the liver and ovarian functions of commercial laying hens are still active in the later laying period, which is the reason for their superior laying performance.

In general, carbohydrate metabolism is more vigorous in the liver of broiler chickens, whereas lipid metabolism is more vigorous in laying hens [28] because broiler chickens need more energy to maintain rapid growth but laying hens need lipids to maintain egg production. Proteomics studies have shown that upregulated DEPs in the liver and ovary of Group G were mainly enriched in pathways related to carbohydrate metabolism, which may be because Guangyuan gray chickens, as a local breed, tend to be used for both meat and eggs. DEPs such as acetyl-coenzyme A synthetase (ACSS1B), s-(hydroxymethyl) glutathione dehydrogenase (ADH4), and hexokinase (HK2), which are involved in carbohydrate metabolism, were identified in the liver of Group G, among which acylphosphatase-1 (*ACYP1*) and *ACSS1B* were also shown to be involved in the regulation of heat stress in chickens [29,30], which may be because Guangyuan gray chickens have a more energetic metabolism and produce more heat, and therefore rely on ACYP1 and ACSS1B to reduce the apoptosis of liver cells in hot environments [29]. In the ovary, glucose-6-phosphatase (G6PC3) [31] and ADH4 have been identified as being involved in carbohydrate metabolism. Notably, ADH4, a unique protein of Group G involved in the catabolism of a variety of substrates such as ethanol, retinol, steroids, and lipids [32,33], has been demonstrated to be a key gene for energy metabolism in cancer cells [34] and can be critical in energy metabolism in the liver and ovary in Group G. Among the DEPs identified in the liver of Group H, only one DEP was enriched in a unique pathway related to carbohydrate metabolism. However, propionate-CoA ligase (ACSS3), ribokinase (RBKS), ATP-citrate synthase (RCJMB04_6f14), and hydroxymethylglutaryl-CoA synthase (HMGCS1) were identified to be involved in carbohydrate metabolism and were found in the ovary of Group H, of which *ACSS3* was also found to be involved in the regulation of heat stress [35] and *HMGCS1* is also involved in the degradation of amino acids and cholesterol [36], which may, to some extent, alleviate fatty liver syndrome in Hy-Line gray chickens.

Notably, in addition to carbohydrate and lipid metabolism, DEPs were enriched in pathways related to amino acid metabolism in Groups G and H. Among them, DEPs such as pyrroline-5-carboxylate reductase (PYCR1), ALDH18A1, aspartate aminotransferase, and cytoplasmic (GOT1) were identified to be significantly enriched in the arginine biosynthesis pathway in the liver of Group G, which is able to inhibit fatty acid synthesis and promote fatty acid β-oxidation [37], which may interfere with the synthesis of substances such as TG and TC in the liver of Guangyuan gray chicken.

On the other hand, most of the upregulated DEPs in the liver and ovary of Group H were enriched in pathways associated with lipid metabolism. No DEPs were identified in the ovary of Group G that were enriched in unique pathways related to lipid metabolism, but ADH4, beta-2-microglobulin (B2M), delta-1-pyrroline-5-carboxylate synthase (ALDH18A1), and 15-oxoprostaglandin 13-reductase (PTGR2), which are involved in lipid metabolism, were identified in the liver. Among them, ADH4 is involved in fatty acid degradation, whereas the overexpression of *PTGR2* suppressed *PPARG*’s transcriptional activity and inhibited lipid synthesis. ALDH18A1 was also a unique protein found in Group G. The loss of function of ALDH18A1 was found to be associated with larger lipid droplets and was positively correlated with leanness in chickens [38], which was consistent with our findings. On the other hand, DEPs such as fatty acid desaturase 1 (FADS1), cytochrome P450 1A4 (CYP1A4), and tryptophan 2,3-dioxygenase (TDO2) were identified in the liver of Group H. These are involved in lipid metabolism, of which CYP1A4 is a protein unique to Group H involved in the metabolism of a variety of compounds, including substances such as steroids and fatty acids [39]. It was identified that steroid sulfatase (STS), glucosylceramidase (LOC107050229), and phospholipase A2 Group XV (PLA2G15) were all involved in the metabolism of functional lipids in the ovary of group H. Among them, *STS* was identified as a key gene for the secretion of reproductive hormones [40]. These three DEPs were unique proteins of group H and may be associated with the high egg production of Hy-Line gray hens at the end of the laying period.

In metabolomic studies, the functional enrichment of DMs was consistent with that of the DEPs. The most abundant metabolites identified in the liver and ovary were organic acids and their derivatives, and lipids and lipid-like molecules, respectively. The involvement of pyruvate, Dl-a-hydroxybutyric acid, L-arabinono-1, 4-lactone, and glycolate in carbohydrate metabolism was identified in the liver and ovary of Group G. Among them, 4-lactone and glycolate have been have been discovered to be critical in fighting oxidative stress [41]. 2-Dehydro-3-deoxy-D-gluconate and D-sorbitol were identified in the ovary of Group H and are involved in carbohydrate metabolism. Sorbitol catalyzes the conversion of fructose in the liver. Excess fructose increases liver stress and contributes to the accumulation of fat [42]. Additionally, the more abundant lipids and lipid-like molecules in the liver of Group G were o-phosphoethanolamine and taurocholate; taurocholate is involved in the biosynthesis of primary bile acid, and bile acid can promote the digestion and absorption of fat [43].

The more abundant lipid and lipid-like molecules in the liver of Group H were glyceric acid, palmitic acid, cis-9-palmitoleic acid, and linolenic acid; the most abundant in the ovary was chenodeoxycholate. A former study found that the liver of laying hens with fatty liver hemorrhage syndrome (FLHS) contained high levels of beta-hydroxybutyric acid, oleic acid, palmitoleic acid, glutamic acid, and other metabolites, which could be used as biomarkers for diagnosing disease [44]. These results were consistent with our findings. Therefore, although Hy-Line gray hens performed better in terms of egg production than Guangyuan gray chickens, prolonged overlaying also caused liver lesions in Hy-Line gray hens. It is also noteworthy that DMs identified in the liver and ovary of Group H were also significantly enriched in the ABC transporter pathway, suggesting the higher transmembrane transport capacity for amino acids, sugars, and other nutrients in Hy-Line gray hens [45]. Moreover, 1-aminocyclopropanecarboxylic acid was identified in the ovary of Group H, which is involved in the synthesis of methionine, a major component of apolipoproteins that significantly increases serum TG levels in laying hens and is important for yolk deposition [46]. In summary, the results of this study suggest that DEPs and DMs involved in the metabolism of carbohydrates, lipids, and amino acids in the liver and ovary of local breeds of chicken and commercial chickens may be responsible for the differences in egg production performance between them in the later laying period.

## 4. Materials and Methods

### 4.1. Birds and Sample Collection

All animal experiments were carried out in accordance with the relevant regulations formulated by the Experimental Animal Operation Standards and Welfare Management Committee of Sichuan Agricultural University (approval no. DKY2021202030).

The chickens used in this experiment were reared by Sichuan Tianguan Agricultural and Animal Husbandry Co., Ltd. (Guangyuan, China). One hundred 1-day-old female chicks were reared, with fifty each of Guangyuan gray chicks and Hy-Line gray chicks. At six weeks of age, each breed was divided into 10 groups, with replicates in each group, and reared in a single cage. At 55 weeks of age, 10 healthy chickens with a similar bodyweight were randomly selected from each of the two breeds (Group G and Group H). Blood was obtained from the pterygoid vein, followed by weighing and euthanizing the chickens by neck dislocation. Liver and ovarian tissues were collected, the livers were weighed, and the pre-grade follicles and grade follicles were counted. The liver and ovary tissues were separated into two sections. One section was immersed in a 4% paraformaldehyde fixative (Sigma-Aldrich, St. Louis, MO, USA) for histological observations, and the other was rapidly refrigerated in liquid nitrogen and kept at −80 °C for later analysis.

### 4.2. Protein Extraction, Digestion and Quality Control

Six liver and ovary samples were collected from each breed for protein profiling. Three replicate groups were established for each of the different tissues of each breed, and each group consisted of a mixed sample of tissues from two chickens of the same breed. An SDT buffer (4% SDS, 1 mM DTT) was used for extracting protein under a high pH. The protein quantification was conducted with the BCA Kit (Bio-Rad, Hercules, CA, USA). Protein digestion followed the Filter-Aided Sample Preparation (FASP) procedure [47]. Each sample initially contained 200 µg of protein, which was mixed with 30 µL of the SDT buffer comprising 4% SDS, 100 mM DTT, and 150 mM Tris-HCl at pH 8.0. We then removed small molecules by treating the samples with a UA buffer containing 8 M urea. Then, 100 μL of iodoacetamide was added, and the samples were incubated in the dark for 30 min. The filters were washed sequentially with the UA buffer and a 25 mM NH_4_HCO_3_ buffer. Next, 4 μg of trypsin (Promega, Madison, WI, USA) was added to a protein/enzyme ratio of 50:1, and the protein suspension was digested at 37 °C overnight, then, the peptides in the filtrate were collected. The digested peptides from each sample were desalted on C18 cartridges (Empore™ SPE Cartridges C18 (standard density) on a bed with an inner diameter of 7 mm and a volume of 3 mL, Sigma-Aldrich, St. Louis, MO, USA), then, they were concentrated by vacuum centrifugation and reconstituted in 40 µL of 0.1% formic acid. Finally, the peptide content was calculated at 280 nm.

A quality control of the protein extraction process was performed using SDS-PAGE. Briefly, 20 µg of protein from each sample was mixed with the loading buffer diluted fivefold and heated for 5 min. The proteins were then separated on a 12.5% SDS-PAGE gel. Finally, we visualized the protein bands by staining with Coomassie Blue R-250.

### 4.3. LC-MS/MS Analysis of Proteins

In this study, a 4D label-free approach and shotgun approach were used for the proteomics experiments, i.e., ion mobility separation was added to the original three dimensions of proteomic separation (retention time, *m*/*z*, and intensity) [48]. The samples were separated using the HPLC liquid phase system Easy nLC, loaded with 400 ng of peptides per run. The peptides were loaded onto a reverse-phase trap column (Acclaim PepMap100, 100 μm × 2 cm, nanoViper C18, Thermo Fisher Scientific, Waltham, MA, USA) connected to the C18 reverse-phase analytical column (Easy Column; 10 cm long, 75 μm inner diameter, 3 μm resin, Thermo Fisher Scientific, Waltham, MA, USA) in Buffer A (0.1% formic acid). The separation of peptides was achieved using a linear gradient of Buffer B (84% acetonitrile and 0.1% formic acid) at a flow rate of 300 nL/min. Then, we conducted the LC-MS/MS analysis using a timsTOF Pro mass spectrometer (Bruker Daltonics, Billerica, MA, USA) coupled with Nanoelute equipment (Bruker Daltonics, Billerica, MA, USA). The mass spectrometer in this study was operated in positive ion mode and the data acquisition mode was the DDA-PASEF mode. The mass range of ion mobility for mass spectrometry acquisition was 100–1700 (*m*/*z*) and 0.6–1.6 (1/k0). Additionally, 10 cycles of PASEF MS/MS with a target intensity of 1.5 k and a threshold of 2500 were performed. To prevent repeated analysis, active exclusion with a release time of 15 s was enabled.

### 4.4. Bioinformatic Analysis

The NR databases (GenBank, Refseq, SwissProt, etc.) and the UniProt databases (SwissProt, TrEmbl) were used for identifying proteins in this study. MaxQuant software 1.6.14 was used for quantitative analysis and the QC of the protein mass spectrometry data [49]. In the analysis, the raw data from the mass spectrometry analysis were entered and the relevant parameters were set to check the database for identification and quantitative analysis. The following configurations were used: a maximum tolerance of 2 missed sites was allowed, with primary and secondary ion mass tolerances set at ±6 ppm and 20 ppm. Moreover, carbamidomethyl (C) and oxidation (M) were designated as fixed and variable modifications, respectively. The database pattern used to calculate the false discovery rate (FDR) was reversed, with a peptide FDR of ≤0.01 and a protein FDR of ≤0.01 as the screening criteria. A quantitative comparison of the proteins between groups was performed using LFQ, and the main algorithm was applied after pairwise correction of the peptide and protein multiples. FC > 2 or FC < 0.50, with *p* < 0.05, was used to screen DEPs that were significantly up- and downregulated between the groups, and volcano plots were generated using the volcano 3D R package (R 4.2.2). We utilized the Complexheatmap R 4.2.2 (R Foundation for Statistical Computing, Vienna, Austria), Cluster 3.0 (Michelson Laboratories, Tokyo, Japan), and Java Treeview software 3.0 (Eisen Lab at Stanford University, Stanford, CA, USA) for sample classification and protein expression level analysis. The classification was performed using the Euclidean distance algorithm and average linkage for clustering. Hierarchical clustering heatmaps were generated based on significant changes in protein expression levels (FC > 2, *p* < 0.05). The DEPs were then functionally annotated with Gene Ontology (GO) terms using Blast2Go software (https://www.blast2go.com, accessed on 20 April 2023, BioBam Bioinformatics, Valencia, Spain), and Kyoto Encyclopedia of Genes and Genomes (KEGG) annotation was performed on the DEPs using KAAS software (https://www.genome.jp/tools/kaas, accessed on 20 April 2023, Kanehisa Laboratories at Kyoto University, Kyoto, Japan). In addition, unique proteins were obtained by using the screening criterion that half or more than half of the samples in one group were not null and all data in the other group were null.

### 4.5. Metabolite Extraction

Six liver and six ovarian samples from each of the two breeds were selected for metabolic profile analysis between the two breeds. After slowly thawing the samples, 5 mg of each sample was added to a pre-cooled solution of methanol, acetonitrile, and water in a ratio of 2:2:1. Then, we vortexed the mixture and treated it ultrasonically at a low temperature for 30 min, followed by ultrasonic treatment at −20 °C for 10 min, and we finally centrifuged the sample at 14,000× *g* at 4 °C for 20 min. The supernatant was then dried and 100 μL of an aqueous solution of acetonitrile was added. We centrifuged the mixture at 14,000× *g* for 15 min and collected the supernatant for ultra-high pressure liquid chromatograph (UHPLC)-MS/MS analysis. The temperature was maintained at 4 °C throughout the experiment.

### 4.6. UHPLC-MS/MS Analysis and Data Analysis

The samples were separated on a Vanquish LC UHPLC (Thermo Fisher Scientific, Waltham, MA, USA) with a HILIC column (ACQUITY UPLC BEH Amide, 1.7 μm, 2.1 mm × 100 mm, Waters, Milford, MA, USA). The detection conditions were as follows: column temperature, 25 °C; flow rate, 300 μL/min. The composition of Mobile Phase A was composed of water, 25 mM acetic acid, and 25 mM ammonia; Phase B was acetonitrile. By gradient elution, within 17 min, B was linearly changed from 98% to 2%, then to 98%, and finally returned to and maintained at 98%. QC samples were inserted in the sample queue to ensure the system stability and data reliability. Metabolite spectra were obtained using a Q Exactive Orbitrap mass spectrometer (Thermo Fisher Scientific, Waltham, MA, USA). The positive and negative ion modes of electric spray ionization (ESI) were used for primary and secondary mass spectra acquisition, with the following ESI settings: Gas1 and Gas2: 60, CUR: 30 psi, ion source temperature: 600 °C, ISVF: ±5500 V. Mass spectra were acquired in DDA mode with secondary spectra obtained by segmented acquisition. Primary and secondary spectra were acquired with the following settings: mass-to-charge ratio detection ranges of 80–1200 Da and 70–1200 Da, resolutions of 60,000 and 30,000, cumulative scan times of 100 ms and 50 ms, asnd a dynamic exclusion time of 4 s.

The raw data format was converted to mzXML using ProteoWizard. Metabolite data were extracted using XCMS software 3.5.1 (Scripps Research Institute, La Jolla, CA, USA), and a substance match value of 0.7 was obtained by considering retention time, molecular mass (molecular mass error: <25 ppm), and mass spectral match (*m*/*z* < 10 ppm, peak width = c (10, 60), prefilter = c (10, 100)). The obtained data were then identified using an in-house database (Shanghai Applied Protein Technology, Shanghai, China) and verified twice. Based on the standards for metabolite identification, the metabolites in this study were classified as Level 2 or higher [50]. PLS-DA was performed on the metabolites that differed between the groups, and the PLS-DA was corrected by OPLS-DA. FC analysis was performed on all metabolites, including those without identification information, to screen for up- or downregulated DMs on the basis of FC > 1.5 or FC < 0.67, and *p* < 0.05. Significant DMs were screened among DMs with identifying information using OPLS-DA, with variable importance in projection (VIP) > 1 as a criterion, and subsequent analysis was performed. Pearson’s correlation analysis was used to identify the relationship between two breeds. In addition, we used Blast2GO and KAAS to perform GO and KEGG pathway annotation of the target metabolite collection. R package (ropls) was used to visualize analyses such as volcano maps and correlation heat maps.

### 4.7. Detection of Serum Biochemical Parameters

The separated serum was centrifuged at 4000 r/min to remove impurities. Then, the hormone levels were tested with an ELISA kit (MyBioSource, Wuhan, China) for chicken VTG, LH, FSH, LEC, and VLDLy according to the manufacturer’s instructions. In addition, TC and TG test kits (Solarbio, Beijing, China) were used to detect the TC and TG levels. The assay was repeated 14 times for three QC samples (1.23 ng/mL, 4.25 ng/mL, and 19.45 ng/mL, respectively) to test the reproducibility of the kit between wells in the assay plate, and the results showed that the CV value of each sample was less than 10% (4.23%, 7.52%, and 7.67%, respectively), indicating that the kit had high precision. The results on the serum hormone levels were visualized using GraphPad Prism 9.0.0 (GraphPad Software, La Jolla, CA, USA).

### 4.8. Oil Red O Staining

Frozen samples were sectioned to 8 μm, fixed in 10% formalin (Solarbio, Beijing, China) for 10 min, and washed. The sections were immersed in 60% isopropanol (Sinopharm, Beijing, China) for 2 min. The sections were stained with an Oil Red O solution (Sangon Biotech, Shanghai, China) for 15 min while protected from the light. The sections were again immersed in 60% isopropanol for 5 s to remove the staining solution and washed again in ice-cold distilled water. The nuclei were stained with Mayer’s hematoxylin (Sangon Biotech, Shanghai, China) for 5 min, washed, dried, and embedded in glycerol gelatin (Xilong Scientific, Shantou, China). The sections were observed under a BX53F inverted microscope (Olympus, Tokyo, Japan) and photographed.

### 4.9. Hematoxylin–Eosin (HE) Staining

The samples were first immobilized in 4% paraformaldehyde (Solarbio, Beijing, China) for 24 h, then immersed in 70%, 80%, 90%, 95%, and 100% ethanol solutions for 30 min to dehydrate the samples. They were then placed in xylene for 2 h to make the samples transparent and embedded in paraffin wax for 3 h. The embedded samples were sectioned into 5 μm pieces and immersed in xylene for 20 min to dewax the samples. The sections were then immersed in a series of ethanol solutions from high to low concentrations and finally in distilled water. The sections were stained with a hematoxylin solution (Beyotime, Haimen, China) for 4 min, fractionated in hydrochloric acid and ethanol for 3 s each, rinsed in running water for 1 h, immersed in distilled water for 10 min, and dehydrated in 70% and 90% ethanol solutions for 10 min each, followed by staining with the eosin staining solution for 3 min. The stained sections were dehydrated by immersion in an ethanol solution and then immersed in xylene to make the sections transparent, and they were finally sealed and stained with gum. The sections were sealed with resin, observed under a BX53F inverted microscope (Olympus, Tokyo, Japan), and photographed.

### 4.10. Western Blot Validation of DMs Expression

After tissue collection, proteins were extracted using the Total Tissue Protein Extraction Kit (Servicebio, Wuhan, China) according to the provided instructions. Protein concentrations were determined using the BCA kit (Servicebio, Wuhan, China) and the samples were made consistent. Protein samples (5 μL) were loaded onto SDS-PAGE gels (separation gel: 10%, concentration gel: 5%) along with a 4:1 ratio of protein to sample loading buffer for electrophoresis. The proteins were then moved to the PVDF membranes with the wet transfer device and blocked for 1 h with blocking solution (Beyotime, Haimen, China). The primary antibodies (Appendix A) were co-incubated with target proteins overnight at 4 °C, followed by removal of unbound antibodies using TBST (Servicebio, Wuhan, China). The membranes were then treated with appropriate secondary antibodies, excess secondary antibody was washed off, and the protein bands were visualized using the Ultra Hypersensitive ECL Chemiluminescence Kit (Servicebio, Wuhan, China). Quantification was performed using ChemiScope Analysis software 6200 (CLINX, Shanghai, China).

### 4.11. Statistical Analysis

This study presented all the findings in the form of the mean ± standard deviation (SD). To determine the significance, we conducted a statistical analysis using either one-way ANOVA or an unpaired Student’s *t*-test with the help of SPSS 26.0 software (IBM Corporation, Armonk, NY, USA). The significance levels were defined as * *p* < 0.05, ** *p* < 0.01, and ^ns^ *p* ≥ 0.05.

## 5. Conclusions

In conclusion, in this study, morphological and histological observations of the liver and ovary of local breeds of chicken and commercial chickens in the later laying period revealed that the serum levels of reproductive hormones in Guangyuan gray chickens were significantly lower than those in Hy-Line gray chickens, there were fewer lipid droplets in the livers of the Guangyuan gray chickens, and their ovaries were atrophied. In contrast, the Hy-Line gray chickens had abundant lipid droplets in the liver and showed fatty liver-like lesions, but the ovaries were still active with follicles in all stages. The results of the proteomic and metabolomic analyses showed that the DEPs and DMs in the liver and ovary were mainly involved in carbohydrate metabolism, lipid metabolism, and amino acid metabolism. These findings can be used to improve the egg production of local breeds of chicken in the later laying period.

## Figures and Tables

**Figure 1 ijms-24-14394-f001:**
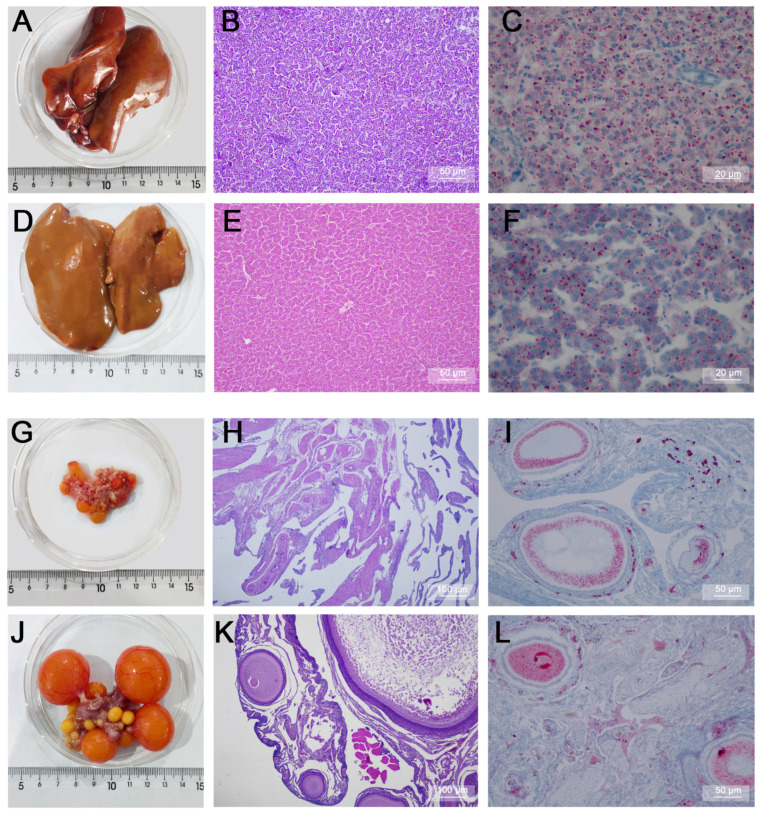
Histological analysis of the liver and ovary in Groups G and H. (**A**) appearance of the liver from Group G. (**B**,**C**) HE result and Oil Red O result of liver staining in Group G. (**D**) appearance of the liver from Group H. (**E**,**F**) HE result and Oil Red O result of liver staining in Group H. (**G**) appearance of the ovaries from Group G. (**H**,**I**) HE result and Oil Red O result of ovary staining in Group G. (**J**) appearance of the ovaries from Group H. (**K**,**L**) HE result and Oil Red O result of ovary staining in Group H. n = 3.

**Figure 2 ijms-24-14394-f002:**
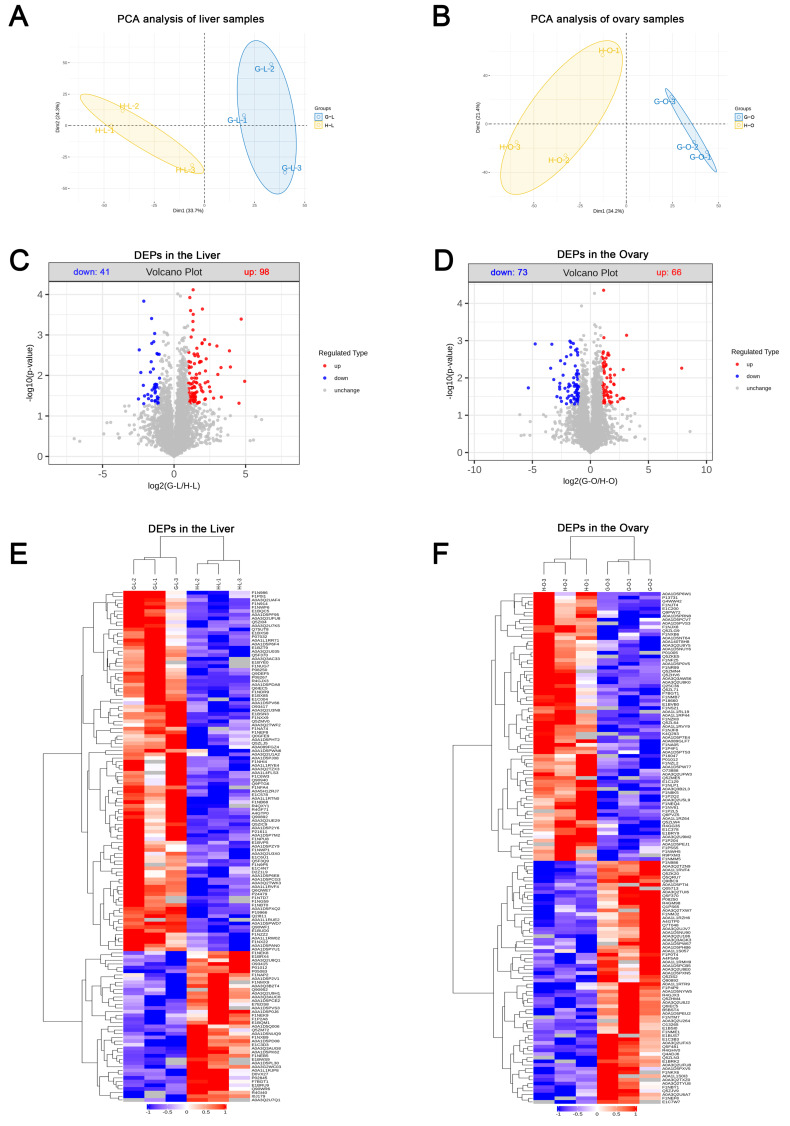
Differential expression analysis of proteins in the liver and ovary between Groups G and H. (**A**,**B**) principal component analysis (PCA) of proteins in the liver and ovary. (**C**,**D**) volcano map of differentially expressed proteins (DEPs) in the liver and ovary. Upregulated DEPs were indicated by red dots, downregulated DEPs by blue dots, and proteins with no significant changes by gray dots. (**E**,**F**) hierarchical clustering analysis of DEPs in the liver and ovary. Red and blue regions indicate significantly upregulated or downregulated proteins, respectively; gray regions indicate no quantitation information.

**Figure 3 ijms-24-14394-f003:**
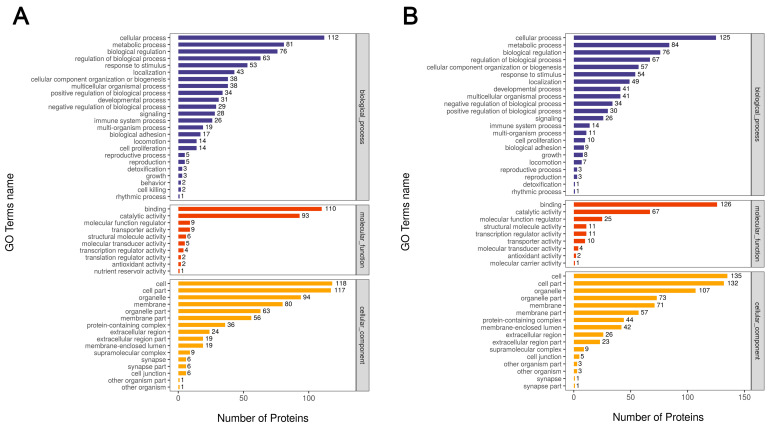
Bar graphs of the GO annotations of the DEPs. (**A**) GO functional annotations of DEPs in the liver including biological processes (BP), cellular components (CC), and molecular functions (MF). The *y*-axis indicates the number of proteins. (**B**) GO functional annotations of DEPs in the ovary.

**Figure 4 ijms-24-14394-f004:**
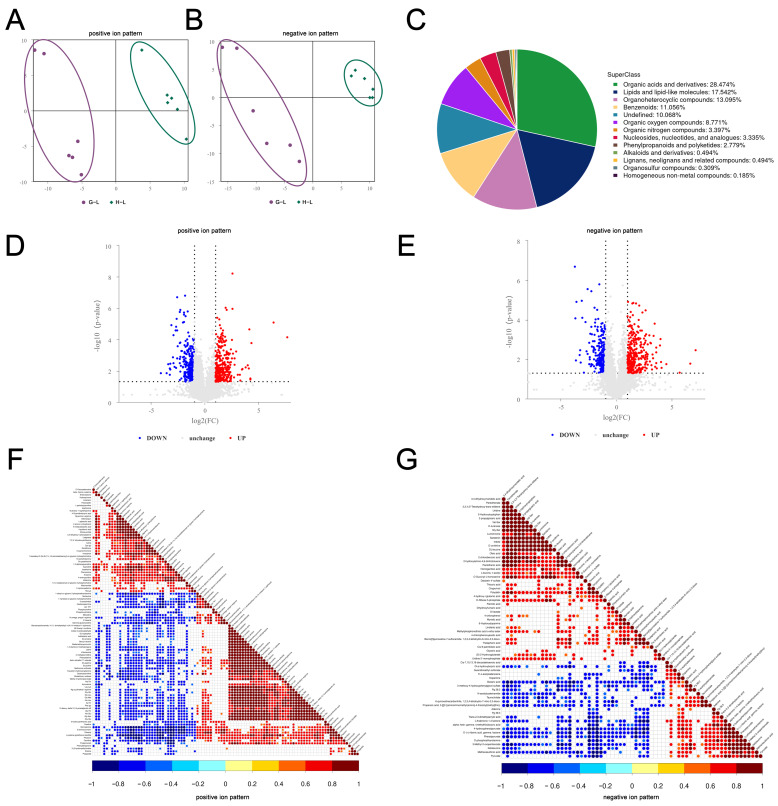
The patterns of differential metabolites (DMs) in the liver between Groups G and H were examined in the positive and negative ion modes. (**A**,**B**) PLS-DA distribution of 12 samples in the positive and negative ion modes. (**C**) the chemical classification of all liver metabolites was determined. (**D**,**E**) volcano plots displayed the DMs identified in the positive and negative ion modes. Upregulated DMs were indicated by red dots, downregulated DMs by blue dots, and metabolites with no significant changes by gray dots. (**F**,**G**) heat maps depicted correlations in the positive and negative ion modes, where red represented positive associations, blue represented negative associations, and white represented non-significant associations.

**Figure 5 ijms-24-14394-f005:**
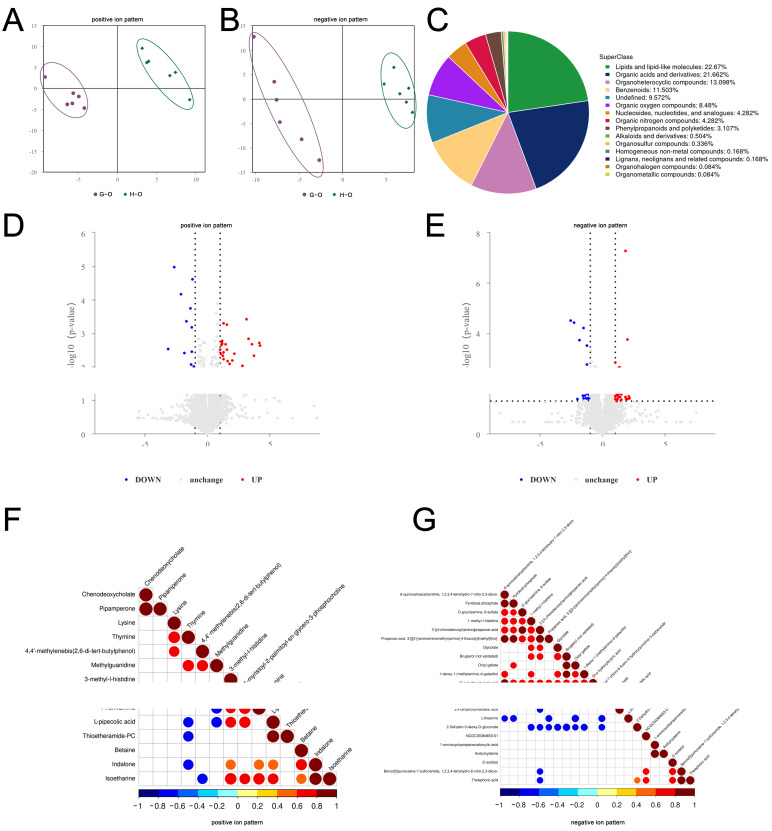
The patterns of DMs in the ovary between Groups G and H were examined in the positive and negative ion modes. (**A**,**B**) PLS-DA distribution of 12 samples in the positive and negative ion modes. (**C**) the chemical classification of all ovary metabolites was determined. (**D**,**E**) volcano plots displayed the DMs identified in the positive and negative ion modes. Upregulated DMs were indicated by red dots, downregulated DMs by blue dots, and metabolites with no significant changes by gray dots. (**F**,**G**) heat maps depicted correlations in the positive and negative ion modes, where red represented positive associations, blue represented negative associations, and white represented non-significant associations.

**Figure 6 ijms-24-14394-f006:**
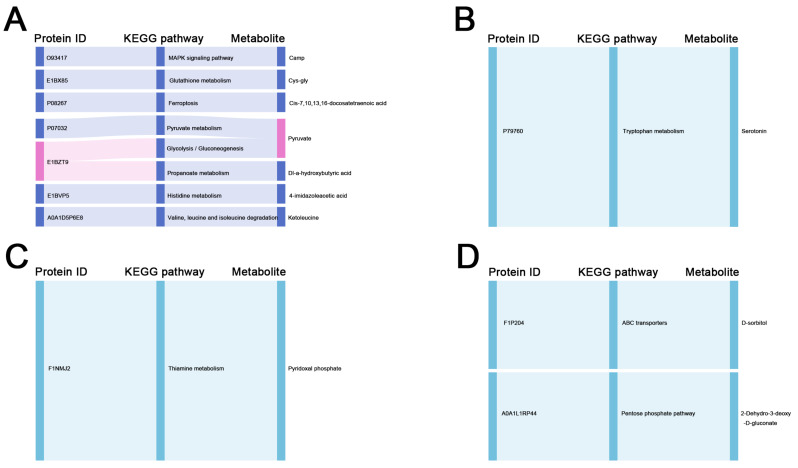
Comprehensive analysis of the KEGG pathways of DEPs and DMs in the liver and ovary between Groups G and H. (**A**,**B**) Sankey diagram of the KEGG pathways of DEPs and DMs in the liver of Groups G and H. (**C**,**D**) Sankey diagram of the KEGG pathways of DEPs and DMs in the ovary of Groups G and H.

**Figure 7 ijms-24-14394-f007:**
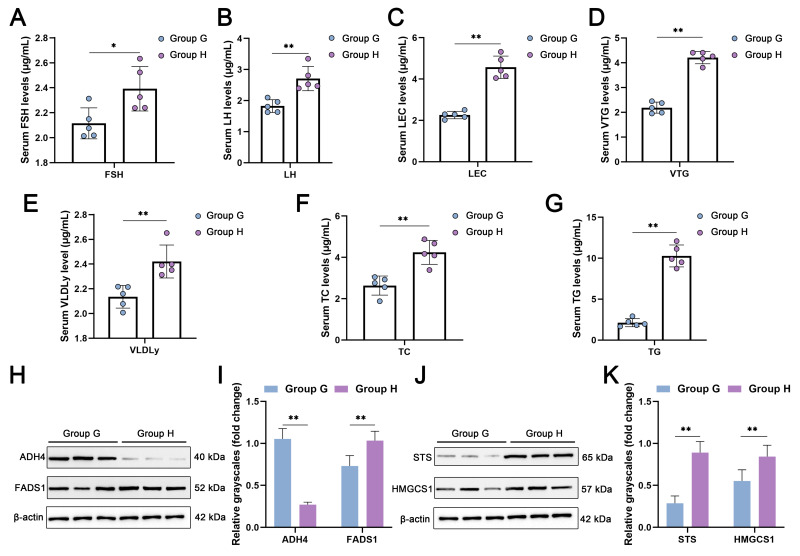
Validation of blood biochemical indicators and protein expression of DEPs found in Groups G and H. (**A**) follicle stimulating hormone (FSH) levels in serum. (**B**) luteinizing hormone (LH) levels in serum. (**C**) lecithin (LEC) levels in serum. (**D**) vitellogenin (VTG) levels in serum. (**E**) very-low-density lipoprotein (VLDLy) levels in serum. (**F**) total cholesterol (TC) levels in serum. (**G**) triglyceride (TG) levels in serum. (**H**,**I**) protein expression of DEPs (ADH4 and FADS1) enriched in carbohydrate and lipid metabolic pathways in the liver. (**J**,**K**) protein levels of DEPs (STS and HMGCS1) enriched in carbohydrate and lipid metabolic pathways in the ovary. All results are presented as the mean ± SD. n = 3. * *p* < 0.05; ** *p* < 0.01.

**Table 1 ijms-24-14394-t001:** Measurements of liver and ovary indices at 55 weeks in Groups G and H.

Groups	Body Weight (g)	Liver Weight (g)	Number of F1–F6 Follicles	Number of YF	Number of WF	Oviduct Length (mm)
G	1611.00 ± 194.44	29.89 ± 3.06	1.50 ± 2.12	12.80 ± 3.39	17.70 ± 5.73	43.25 ± 10.59
H	1858.00 ± 185.34 **	36.21 ± 1.13 **	5.70 ± 0.48 **	16.80 ± 3.15 *	14.20 ± 2.85	76.32 ± 6.68 **

YF, yellow follicles, diameter 5–10 mm; WF, white follicles, diameter < 5 mm. All results are presented as the mean ± SD. n = 10. * *p* < 0.05, ** *p* < 0.01.

**Table 2 ijms-24-14394-t002:** Pathways of upregulated differentially expressed proteins (DEPs) enriched in the liver between Groups G and H.

Group	Classification	KEGG Pathway	Protein ID	Name
G	Carbohydrate metabolism	Pyruvate metabolism	sp|P07032|ACYP1_CHICK	ACYP1
		Pyruvate metabolism	tr|E1BZT9|E1BZT9_CHICK	ACSS1B
		Pyruvate metabolism	tr|F1NI89|F1NI89_CHICK	ADH4
		Propanoate metabolism	tr|E1BZT9|E1BZT9_CHICK	ACSS1B
		Glycolysis, gluconeogenesis	tr|E1BZT9|E1BZT9_CHICK	ACSS1B
		Glycolysis, gluconeogenesis	tr|F1NI89|F1NI89_CHICK	ADH4
		Glycolysis, gluconeogenesis	tr|A0A3Q2UE29|A0A3Q2UE29_CHICK	HK2
		Fructose and mannose metabolism	tr|A0A3Q2UE29|A0A3Q2UE29_CHICK	HK2
	Amino acid metabolism	Valine, leucine, and isoleucine degradation	tr|A0A1D5P6E8|A0A1D5P6E8_CHICK	CDH15
		Tyrosine metabolism	tr|F1NI89|F1NI89_CHICK	ADH4
		Histidine metabolism	tr|E1BVP5|E1BVP5_CHICK	ASPA
		Arginine and proline metabolism	tr|F1N914|F1N914_CHICK	PYCR1
		Arginine and proline metabolism	tr|E1BSN3|E1BSN3_CHICK	L3HYPDH
		Arginine and proline metabolism	tr|A0A1L1RR71|A0A1L1RR71_CHICK	ALDH18A1
	Lipid metabolism	Fatty acid degradation	tr|F1NI89|F1NI89_CHICK	ADH4
		Fatty acid biosynthesis	tr|A0A1L1RR71|A0A1L1RR71_CHICK	ALDH18A1
		Fatty acid biosynthesis	sp|P21611|B2MG_CHICK	B2M
		Arachidonic acid metabolism	tr|Q5F370|Q5F370_CHICK	PTGR2
H	Lipid metabolism	Cutin, suberin, and wax biosynthesis	sp|Q5ZM72|FACR1_CHICK	FAR1
		Biosynthesis of unsaturated fatty acids	tr|E7EDS8|E7EDS8_CHICK	FADS1
	Amino acid metabolism	Tryptophan metabolism	sp|P79760|CP1A4_CHICK	CYP1A4
		Tryptophan metabolism	tr|F1NEK9|F1NEK9_CHICK	ACMSD
		Tryptophan metabolism	tr|A0A1D5P0J6|A0A1D5P0J6_CHICK	TDO2

**Table 3 ijms-24-14394-t003:** Pathways of upregulated DEPs enriched in the ovary between Groups G and H.

Group	Classification	KEGG Pathway	Protein ID	Name
G	Amino acid metabolism	Tyrosine metabolism	tr|F1NTM7|F1NTM7_CHICK	GOT1
		Phenylalanine, tyrosine, and tryptophan biosynthesis	tr|F1NTM7|F1NTM7_CHICK	GOT1
		Cysteine and methionine metabolism	tr|F1NTM7|F1NTM7_CHICK	GOT1
		Arginine biosynthesis	tr|F1NTM7|F1NTM7_CHICK	GOT1
		Arginine and proline metabolism	tr|F1NTM7|F1NTM7_CHICK	GOT1
		Alanine, aspartate, and glutamate metabolism	tr|F1NTM7|F1NTM7_CHICK	GOT1
		Glycolysis, tyrosine metabolism	tr|F1NI89|F1NI89_CHICK	ADH4
	Carbohydrate metabolism	Starch and sucrose metabolism	tr|A0A3Q2U8J2|A0A3Q2U8J2_CHICK	G6PC3
		Pyruvate metabolism	tr|F1NI89|F1NI89_CHICK	ADH4
		Glycolysis, gluconeogenesis	tr|A0A3Q2U8J2|A0A3Q2U8J2_CHICK	G6PC3
		Glycolysis, gluconeogenesis	tr|F1NI89|F1NI89_CHICK	ADH4
		Galactose metabolism	tr|A0A3Q2U8J2|A0A3Q2U8J2_CHICK	G6PC3
H	Amino acid metabolism	Valine, leucine, and isoleucine degradation	tr|A0A1L1RZ64|A0A1L1RZ64_CHICK	HMGCS1
		Tryptophan metabolism	tr|A0A3Q3AW56|A0A3Q3AW56_CHICK	GCDH
		Lysine degradation	tr|A0A3Q3AW56|A0A3Q3AW56_CHICK	GCDH
		Histidine metabolism	tr|E1C378|E1C378_CHICK	HNMT
	Carbohydrate metabolism	Propanoate metabolism	tr|F1P5S5|F1P5S5_CHICK	AC0SS3
		Pentose phosphate pathway	tr|A0A1L1RP44|A0A1L1RP44_CHICK	RBKS
		Citrate cycle (TCA cycle)	tr|Q5F3V2|Q5F3V2_CHICK	RCJMB04_6f14
		Butanoate metabolism	tr|A0A1L1RZ64|A0A1L1RZ64_CHICK	HMGCS1
	Lipid metabolism	Steroid hormone biosynthesis	tr|A0A1D5PEM6|A0A1D5PEM6_CHICK	STS
		Sphingolipid metabolism	tr|A0A1D5NWE6|A0A1D5NWE6_CHICK	LOC107050229
		Glycerophospholipid metabolism	tr|A0A1D5PU31|A0A1D5PU31_CHICK	PLA2G15
	Cell growth and death	Oocyte meiosis	tr|A0A3Q2U5L9|A0A3Q2U5L9_CHICK	RPS6KA1

**Table 4 ijms-24-14394-t004:** Pathways of upregulated differential metabolites (DMs) enriched in the liver between Groups G and H.

Group	Classification	KEGG Pathway	Metabolite
Group G	Amino acid metabolism	Valine, leucine, and isoleucine degradation	Ketoleucine
		Valine, leucine, and isoleucine biosynthesis	Ketoleucine, pyruvate
		Histidine metabolism	4-Imidazoleacetic acid, anserine, histamine, N-acetylhistamine
		Alanine, aspartate, and glutamate metabolism	Pyruvate
		Phenylalanine, tyrosine, and tryptophan biosynthesis	4-Hydroxyphenylpyruvate, phenylpyruvate
		Phenylalanine metabolism	Phenylpyruvate, pyruvate
	Carbohydrate metabolism	Pyruvate metabolism	Pyruvate
		Propanoate metabolism	Dl-a-hydroxybutyric acid
		Pentose and glucuronate interconversions	Pyruvate
		Glycolysis, gluconeogenesis	Pyruvate
		Citrate cycle (TCA cycle)	Pyruvate
		Ascorbate and aldarate metabolism	L-arabinono-1,4-lactone, pyruvate
	Endocrine system	Progesterone-mediated oocyte maturation	Camp
		Insulin signaling pathway	Camp
		GnRH signaling pathway	Camp
	Lipid metabolism	Sphingolipid metabolism	O-phosphoethanolamine
		Primary bile acid biosynthesis	Taurocholate
Group H	Lipid metabolism	Glycerolipid metabolism	Glyceric acid
		Fatty acid elongation	Palmitic acid
		Fatty acid biosynthesis	Cis-9-palmitoleic acid, palmitic acid
		Alpha-linolenic acid metabolism	Linolenic acid
	Amino acid metabolism	Tryptophan metabolism	Serotonin, 3-(2-hydroxyethyl) indole, 3-hydroxyanthranilic acid, N-acetyl-5-hydroxytryptamine

**Table 5 ijms-24-14394-t005:** Pathways of upregulated DMs enriched in the ovary between Groups G and H.

Group	Classification	KEGG Pathway	Metabolite
Group G	Amino acid metabolism	Lysine degradation	L-pipecolic acid
		Histidine metabolism	1-Methyl-l-histidine, 3-methyl-l-histidine
	Carbohydrate metabolism	Propanoate metabolism	Dl-a-hydroxybutyric acid
		Glyoxylate and dicarboxylate metabolism	Glycolate
Group H	Carbohydrate metabolism	Pentose phosphate pathway	2-Dehydro-3-deoxy-D-gluconate
		Pentose and glucuronate interconversions	2-Dehydro-3-deoxy-D-gluconate
		Galactose metabolism	D-sorbitol
		Fructose and mannose metabolism	D-sorbitol
	Amino acid metabolism	Tyrosine metabolism	3,4-Dihydroxymandelic acid
		Cysteine and methionine metabolism	1-Aminocyclopropanecarboxylic acid
	Membrane transport	ABC transporters	D-sorbitol
	Lipid metabolism	Primary bile acid biosynthesis	Chenodeoxycholate

## Data Availability

The mass spectrometry proteomics data have been deposited to the ProteomeXchange Consortium (http://proteomecentral.proteomexchange.org, accessed on 5 September 2023) via the iProX partner repository with the dataset identifier PXD045098. Metabolomics raw data have been deposited in Metabolights with the dataset identifier MTBLS8517.

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
