# Peer review of "Comparative Analysis of Different Proteins and Metabolites in the Liver and Ovary of Local Breeds of Chicken and Commercial Chickens in the Later Laying Period"

_ijms, 2023, doi:10.3390/ijms241814394_

Round 1
Reviewer 1 Report
The manuscript entitled, “Comparative Analysis of Differential Proteins and Metabolites 2
in the Liver and Ovary of Local Breed Chickens and Commercial Chickens at Later Laying Period” implemented proteomics and metabolomics approach to better understand role of liver and ovary in egg formation in hens. The results are not conclusive and not supported by sufficient literature. The manuscript requires proofreading in terms of further English language, rephrasing and typo editing. I cannot recommend this manuscript for publication in its present form.
To be specific, major issues are related to:
Comment1: Author should mention that they used label free approach and single shot approach for proteomics experiments.
Comment2: Author should provide SDS image as supplementary figure as a quality check.
Comment3: As author has used three biological replicated for each categories, author should provide the information of reproducibility between replicates.
Comment4: Author should do validation for few proteins involved in interested pathways using western blot/SRM/MRM. Without validation this manuscript won’t be consider for further review.
Comment5:All figures are poor quality. Author should provide figures with high resolution.
The manuscript requires proofreading in terms of further English language, rephrasing and typo editing.
Reviewer 2 Report
The manuscript ijms-2571525 with the title “Comparative Analysis of Differential Proteins and Metabolites in the Liver and Ovary of Local Breed Chickens and Commercial Chickens at Later Laying Period” investigated the intricacies of liver and ovary dynamics in both local Guangyuan gray and commercial Hy-Line gray chickens, shedding light on the nuanced interplay of metabolic pathways and protein expressions that underlie the contrasting egg production capacities observed between these two groups during their later laying period.
The manuscript is of great importance for proteomics and metabolomics, very well organized, and below are some comments for improving the manuscript:
- Figure 2 (legend), lines 126-127: In the caption should be specified the long name for the biochemical indicators. Example: Follicle-stimulating hormone (FSH) etc.
- Section 2.2: What FDR threshold was applied to the data?
- Figure 3 (A to H): These panels are informative as quality control for used proteomic methods. Maybe it is more appropriate to add them in supplementary materials.
- Figure 3 (I and J): I recommend to the authors to add some text on the I and J panels so that the reader can easily differentiate between the heatmap for the liver and the heatmap for the ovary. I guess that the heatmap from panel I is for the liver and the one from panel J is for the ovary, right? Also, I recommend to the authors to add the full names of the proteins for each row of the heatmap.
- Section 2.3: Why authors performed GO analysis for proteomic data and not a Pathway enrichment analysis (PEA) dedicated for proteins?
- Figure 4: The resolution of the graphics in the figures must be improved to at least 300 dpi. It may help to represent panels A and B one below the other, vertically instead of horizontally as they are now. Also, I recommend to revers OX to OY axis so that GO annotation can be read “without breaking your neck”.
- Figure 5: FDR applied should be specified. Also, in the text, for DEPs and DMs.
- The tools by which the heatmaps /graphs /volcano plots were generated should be mentioned in the appropriate captions for all figures in the manuscript.
- Section 3. Discussion: For a better understanding of the reader, I recommend to the authors to add the long name of the proteins corresponding to the abbreviations in text.
- Line 392: Specify the ratio of trypsin: protein used and the initial amount of digested protein.
- Section 4.3: Detailed acquisition settings for MS method (PASEF) should be specified in order to assure reproducibility.
- Section 4.3: Has MS data been uploaded on a recognized repository platform? If so, please provide access to raw data files acquired by TimsTOF Pro instrument.
- Section 4.3: Out of the 50 individuals X six samples from each organ, how many for LS-MS samples acquisition were used form each group? Have you run technical replicates also?
- Section 4.4: Details about protein identification and library used or how libraries were built should be specified.
- Line 427: “ultrasonic treatment at 14 000 g” or “centrifuged at 14 000 g” What exactly are the authors referring to?
- Section 4.6: The MS method used (SWATH, MRM) and acquisition details should be specified in order to assure reproducibility. Also, how were the metabolites identified and with what settings? What library was used? Please, provide these essential information’s.
- Has MS data from metabolite samples been uploaded on a recognized repository platform? If so, please provide access to raw data files acquired by TT6600 instrument.
The reproducibility of this study is almost zero without providing the details mentioned above.
Reviewer 3 Report
Dear authors,
Thank you for submitting your manuscript on "Comparative Analysis of Differential Proteins and Metabolites in the Liver and Ovary of Local Breed Chickens and Commercial Chickens at Later Laying Period".
Your topic is of extraordinary importance for agricultural companies and farmers alike.
Although a lot of information has been presented in the manuscript, it looks like there was more than one author since the paragraphs of the manuscript strongly differ.
I have the following comments I would like you to address:
1. Uploading raw analysis data and enabling another scientist to reanalyze and scrutinize the data is crucial. You did not upload the data either for proteomics or for metabolomics data. Please upload the raw data on, e.g., PRIDE for sharing. I cannot accept a manuscript without authors sharing the data.
2. The method part for proteomics analysis, page 17/24, line 401 ff., is very poorly written. The description of the Lc-MS/MS method is not acceptable in its current form. Please rewrite the full section and use the proper nomenclature, e.g., describing the LC column (no column details are given), separation instead of "divide", etc. The description of the metabolomics method is significantly better written.
3. Details of the MS method are completely missing. Please update these details and also use the proper nomenclature. What was the mobility range? Did you use the DDA or the DIA approach, etc.?
4. The discussion section needs to be reorganized. A lot of details are presented, and the data is valid. However, I suggest dividing the discussion section into "proteomics' and "metabolomics" in order to get a better overview.
Kind regards
Different sections of the manuscript differ. There are sections written well, and that can be well understood and portions that are poorly written. Please engage a native English speaker or a well-trained English expert to proofread and correct.
Round 2
Reviewer 2 Report
The manuscript has been greatly improved, the authors answered and solved all my comments and suggestions, but following the answers received I have only a minor comment:
- Section 4.3: How many peptides (quantity) were loaded /injected for each run in the LC separation?
I recommend to the authors to add this information in manuscript.
Reviewer 3 Report
Dear Authors,
Thank you for addressing the issues mentioned in mine and the review of my fellow Reviewer.
Please, check again the language (typos).
I have no further comments and can accept the manuscript in its current form.
Kind regards
Dear Editor,
Authors addresses all issues and I have no further comments.
Kind reagards,
Goran Mitulović
Author Response
Dear Reviewer,
Your previous suggestions and questions have been crucial to the improvement of this manuscript, and your patience and care are much appreciated. We have further checked and corrected the spelling of the manuscript. Thank you again!